# Near-Infrared MAO A Inhibitor (NMI) Outperformed FDA-Approved Chemotherapeutic Agents in Brain and Other Cancers: A Bioinformatic Analysis of NCI60 Screening Data

**DOI:** 10.3390/brainsci11101318

**Published:** 2021-10-04

**Authors:** Qianhua Feng, Yuxuan Lian, Yihan Qian, Jean C. Shih

**Affiliations:** 1Department of Pharmacology and Pharmaceutical Sciences, School of Pharmacy, University of Southern California, Los Angeles, CA 90089, USA; qianhuaf@usc.edu (Q.F.); lianyuxu@usc.edu (Y.L.); yihanqia@usc.edu (Y.Q.); 2Department of Integrative Anatomical Sciences, Keck School of Medicine, University of Southern California, Los Angeles, CA 90033, USA; 3Norris Comprehensive Cancer Center, Keck School of Medicine, University of Southern California, Los Angeles, CA 90089, USA

**Keywords:** MAO A, NMI, NCI60 screening, CNS cancer, prostate cancer, NSCLC

## Abstract

Our previous work has shown that monoamine oxidase A (MAO A) is overexpressed in glioma and prostate cancer. Near-infrared dye conjugate MAO A Inhibitor (NMI) inhibited the growth of these cancers. This study investigated the effects of NMI on other cancers by NCI60 screening. Our results showed that 48 out of 59 screened cell lines from nine types of cancer had 100% growth inhibition at 10 μM NMI treatment. The in vitro efficacy of NMI determined by growth inhibition (GI_50_ and TGI) and lethal doses (LC_50_) has been further studied in various cell lines of CNS cancer, prostate cancer, and non-small cell lung cancer (NSCLC), these three cancers showed increased MAO A expression in tumors compared to normal tissues. Based on the waterfall plots and the 3D scatter plot of GI_50_, TGI, and LC_50_ data, NMI showed higher potency to several CNS cancer and NSCLC cell lines than prostate cancer cell lines. In vitro efficacy of NMI outperformed FDA-approved drugs for CNS cancer, prostate cancer, and NSCLC, respectively. The Pairwise Pearson Correlation Coefficient (PCC) showed that NMI has a unique mechanism compared to the existing anticancer drugs. This study shows that NMI is a novel theragnostic drug with high potency and unique mechanisms for brain, prostate, NSCLC, and other cancers.

## 1. Introduction

Monoamine oxidase A (MAO A) is a mitochondrial enzyme that degrades monoamine neurotransmitters and dietary monoamines via oxidative deamination [1,2]. Our previous studies have shown that MAO A expression is increased in glioma [3] and prostate cancer [4]. Mice bearing glioma treated with MAO A inhibitors showed a significant increase in macrophages and TNF-α expression, resulting in the reduction in tumor progression [3]. MAO A expression promotes reactive oxygen species (ROS) production, epithelial-to-mesenchymal transition (EMT), and tumor hypoxia, resulting in prostate tumorigenesis, progression, and metastasis [4].

NMI (near-infrared (NIR) dye conjugated with a MAO A specific inhibitor, clorgyline, specifically targets tumors by activating the HIF1α/OATPs signaling axis in mitochondria [5].

In vitro studies showed that NMI inhibits the invasion of patient-derived glioma cells. In vivo study showed that NMI alone or in combination with temozolomide (TMZ) increases the survival of U251R (TMZ-resistant) tumor-bearing nude mice [3].

NMI reduces prostate cancer growth as shown by in vitro colony formation, migration, and invasion of prostate cancer cells. In vivo studies using prostate cancer animal models showed that NMI reduces the tumor growth, prostate-specific antigen (PSA), and tumor MAO A activity. Thus, NMI may be an effective therapy for prostate cancer [5].

NMI can be visualized by non-invasive imaging, so NMI has a dual function for diagnosis and therapy for glioma and prostate cancer [5]. The biodistribution and pharmacokinetics of NMI have been studied [6].

To further investigate the potency of NMI to other cancers, NCI60 screening is performed and the results are discussed here.

NCI60 screening is provided by the NCI Developmental Therapeutics Program (DTP), a large-scale bioinformatic database used to discover and develop the therapeutic activities and potency of new anticancer drugs. NCI60 screens 60 human tumor cell lines representing nine different cancers including leukemia, NSCLC, colon cancers, CNS cancers, melanoma, ovarian cancers, renal cancers, prostate cancers, and breast cancers. The NCI60 screening data identify and characterize novel agents by growth inhibition (GI_50_ and TGI) or killing of tumor cell lines (LC_50_). The data of the most common compounds are available on the DTP website (https://dtp.cancer.gov, accessed on 20 June 2021). COMPARE is provided on the website, it can be used to perform interactive analysis of screening data for various compounds. The COMPARE analysis searches for the mechanism of action for new agents.

## 2. Materials and Methods

### 2.1. Screening Methodology

The detailed description of the screening methodology is shown on the website https://dtp.cancer.gov/discovery_development/nci-60/methodology.htm (accessed on 20 June 2021). Briefly, each of the 60 cell lines were seeded in 96-well microtiter plates and incubated for one day under appropriate conditions. Then drug with various concentrations was added into each plate for two days. The percentage of cell growth at various drug concentrations was calculated for the determination of GI_50_, TGI, and LC_50_ [7].

### 2.2. MAOA Expression Levels in Cell Lines Used for NCI60

MAO A expression (tumor/normal) in nine different cancers were obtained from two databases. The Cancer RNA-Seq Nexus (CRN) database is an open resource containing coding-transcript/IncRNA expression data of cancers (http://syslab4.nchu.edu.tw/CRN, accessed on 20 June 2021). The European Molecular Biology Laboratory-European Bioinformatics Institute (EMBL-EBI) database provides gene expression levels under various conditions (https://www.ebi.ac.uk/, accessed on 20 June 2021).

### 2.3. Min-Max Normalization

GI_50_, TGI, and LC_50_ are three separate but correlated parameters. To consider these 3 parameters together, min–max normalization was used to calculate normalized GI_50_, TGI, and LC_50_ for each drug into a range (0, 1) by an equation which was identified as [8]:
(1)$$x\prime =\frac{x-\min\left(x\right)}{\max\left(x\right)-\min\left(x\right)}$$where x was the original value of endpoints, min and max were the minimum and maximum values in the set of observed endpoint values; score 0 meant the lowest concentration required for each parameter, and score 1 meant the highest. Take GI_50_ as an example: x was GI_50_ of each drug, max(x) was the maximum GI_50_ among all CNS/prostate/NSCLC cancer drugs, and min(x) was the minimum GI_50_ among them. TGI and LC_50_ were calculated and normalized in the same way. Then three parameters were added up to calculate the cumulative score (score 0 meant the highest efficacy, score 3 meant the lowest efficacy).

### 2.4. COMPARE

Analyses and the Pairwise Pearson correlation coefficients were calculated by using a commercial statistical package procedure (Statistical Analysis System, RRID:SCR_008567). A PCC of 1.0 identifies a perfect match in patterns, a PCC of -1.0 denotes a perfect mirror image, whereas a PCC of 0 means there is no correlation between the two patterns. High PCC (>0.8, shown in yellow, orange, or red in the matrix COMPARE figure) means two drugs have similar patterns, which indicates that they have similar mechanisms in cancer therapy [9].

### 2.5. Statistical Analysis

The 3D scatter plot of NMI combining GI_50_, TGI, and LC_50_ was made using R (version 3.6.1).

## 3. Results

As shown in Figure 1, there are 59 cell lines representing nine different cancers used for NCI60 analysis. One cell line was dyed. NMI inhibition of cell growth in each cell line was determined by five-dose screening, from 10^−4^ to 10^−8^ M. At 10 μM, 48 cell lines had 100% growth inhibition and nine cell lines had 50% growth inhibition.

To further investigate the potency of NMI to these nine different cancers, NCI60 screening data of GI_50_, TGI and LC_50_ values of NMI were arranged and made into waterfall plots (Figure 2).

The drug efficacy of NMI was determined by the values of these three parameters, the higher drug efficacy required a lower concentration in GI_50_, TGI, and LC_50_. NMI was most potent to the cell line at the top, and the potency of NMI decreased in cell lines below. Although leukemia cell lines showed great potency to NMI in GI_50_ (at the top of the waterfall plot), they performed worse in TGI and were the worst in LC_50_; overall, they were not potent to NMI treatment. Thus, all three parameters should be considered to evaluate the efficacy of drugs. Comparing the efficacy of NMI in the cell lines of CNS cancer, five out of six CNS cancer cell lines (SF-295, SF-539, SNB-19, SNB-75, U251) were higher than the prostate cancer cell line DU-145, which was studied experimentally in our laboratory.

Based on the database search from CRN and EMBL-EBI, CNS cancers have higher MAO A expression in tumors than that in normal tissues (Table 1). The similar results of MAO A expression are also shown in prostate and NSCLC cancers. The MAO A expressions were measured by the RNA-seq in both databases [10,11]. Thus, the cell lines from these three types of cancer were used for further studies.

Based on the CRN database, MAO A expression increased 1.34-fold in tumors compared to normal tissues in CNS cancer, 1.51-fold in prostate cancer, and 2.29-fold in NSCLC, respectively. Similar results were shown in the EMBL-EBI database. In contrast, colon cancer, melanoma, ovarian cancer, renal cancer, and breast cancer have decreased levels of MAO A expression than that in normal tissues [10,11].

For better visualization of the analysis, R was used to correlate GI_50_, TGI, and LC_50_ of NMI into a 3D figure (Figure 3).

The cell line plotted at the top right of the box was the most potent, and the one at the bottom left was the least potent. Figure 3 showed that the dots of four cell lines, CNS cancer (SF-539, U251, colored in yellow), and NSCLC (HOP-92, NCI-H522, colored in brown), plotted at the top right corner of the 3D scatter plot with their names highlighted in red, exhibited higher potency to NMI than both prostate cancer cell lines (colored in magenta).

The results of waterfall plots and the 3D scatter plot together showed that NMI was more efficient to CNS cancer cell lines (SF-539, U251), with (GI_50_, TGI, LC_50_) of (10^−5.92^, 10^−5.57^, 10^−5.22^) and (10^−6.13^, 10^−5.68^, 10^−5.30^); NSCLC cell lines (HOP-92, NCI-H522), with (GI_50_, TGI, LC_50_) of (10^−5.80^, 10^−5.45^, 10^−5.10^) and (10^−5.94^, 10^−5.48^, 10^−5.02^), and prostate cancer cell lines (PC-3, DU-145), with (GI_50_, TGI, LC_50_) of (10^−5.77^, 10^−5.40^, 10^−5.02^) and (10^−5.52^, 10^−4.96^, 10^−4.46^), compared with other cell lines of CNS cancer and NSCLC in NCI60 analysis. Thus, NMI showed significant efficacy to several cell lines in NCI60 (Figure 2 and Figure 3).

Next, the efficacy of NMI was compared with anticancer drugs approved by the FDA for CNS cancer, prostate cancer, and NSCLC. To date, there are six CNS cancer drugs approved by the FDA; four of them were screened by NCI60 (Temozolomide, Lomustine, Carmustine, and Everolimus). The mechanism and indication of these drugs were shown in Appendix A.

Efficacy determined by GI_50_, TGI, and LC_50_ of NMI and CNS cancer drugs were shown in Figure 4 (Figure 4A: GI_50_, Figure 4B: TGI, Figure 4C: LC_50_). The GI_50_ values showed NMI had the highest efficacy compared with four FDA-approved CNS cancer drugs in six CNS cancer cell lines (Figure 4A). The TGI values showed NMI had the greatest efficacy among all FDA-approved CNS cancer drugs in all cell lines (Figure 4B). NMI had the lowest LC_50_ value compared to other CNS cancer drugs in SF-295, SF-539, SNB-19, SNB-75, and U251. In the SF-246, the LC_50_ efficacy of NMI was lower than everolimus and similar to temozolomide (Figure 4C). Based on the GI_50_, TGI, and LC_50_ values, NMI had higher efficacy than most of these four CNS cancer drugs.

The min–max normalization of GI_50_, TGI, and LC_50_ and their cumulative scores for NMI and FDA-approved drugs in NCI60 CNS cancer, prostate cancer, and NSCLC cell lines were shown in Appendix A. As shown in Figure 4D, NMI had the best inhibition results (lowest cumulative score) in CNS cancer cell lines (SF-295, SF-539, SNB-19, SNB-75, and U251), and the second-lowest cumulative score in SF-268. Thus, NMI outperformed other existing drugs’ in vitro efficacy for the treatment of CNS cancer.

The same analysis of NMI and existing cancer drugs was performed similarly on prostate cancer and the results are shown in the supplementary data (Appendix A).

There are eighteen prostate cancer drugs approved by the FDA; eight of them were screened by NCI60 (Rucaparib, Olaparib, Enzalutamide, Abiraterone, Cabazitaxel, Docetaxel, Leuprolide, and Mitoxantrone). The mechanism and indication of these drugs were shown in Appendix A.

Efficacy determined by GI_50_, TGI, and LC_50_ of NMI and prostate cancer drugs were shown in Appendix A (Appendix A: GI_50_, Appendix A: TGI, Appendix A: LC_50_). The result of GI_50_ showed that NMI was more efficient compared with many FDA-approved prostate cancer drugs in both cell lines (Appendix A). The TGI of NMI exhibited higher efficacy among all the FDA-approved prostate cancer drugs except mitoxantrone in the PC-3 cell line, and cabazitaxel and mitoxantrone in the DU-145 cell line (Appendix A). The LC_50_ of NMI showed the second-highest efficacy compared to other FDA-approved prostate cancer drugs in both cell lines (Appendix A). Based on the GI_50_, TGI, and LC_50_ values, NMI displayed better efficacy than most of the existing prostate cancer drugs as a potent drug candidate for prostate cancer.

By combining GI_50_, TGI, and LC_50_, the cumulative scores of NMI and six prostate cancer drugs were shown in Appendix A. NMI had the second-lowest cumulative score, showing that NMI had better inhibition results in prostate cancer cell lines (PC-3 and DU-145) than these FDA-approved drugs except Mitoxantrone. Therefore, NMI had great potency for treating prostate cancer.

Similarly, the potency of NMI and FDA-approved NSCLC drugs were compared and the results are shown in Appendix A.

There are thirty-four NSCLC cancer drugs approved by the FDA; fifteen of them were screened by NCI60 (Alectinib, Pemetrexed, Carboplatin, Crizotinib, Docetaxel, Doxorubicin, Erlotinib, Everolimus, Gefitinib, Lorlatinib, Mechlorethamine, Trametinib, Methotrexate, Paclitaxel, Vinorelbine). The mechanism and indication of these drugs were shown in Appendix A.

Appendix A showed the comparison of GI_50_ values of NMI with fifteen FDA-approved NSCLC drugs in nine NSCLC cell lines. Appendix A showed TGI values and Appendix A showed LC_50_ values. The cumulative scores of NMI and 15 NSCLC drugs were shown in Appendix A. Taken together, these results showed that NMI had better inhibition of cancer growth (a lower cumulative score) than most of these drugs in NSCLC cell lines, indicating that NMI is a potential anticancer drug for NSCLC.

Next, we studied the mechanism of NMI. We used the COMPARE algorithm from DTP’s anticancer screening program to calculate the Pairwise Pearson Correlation Coefficient (PCC) and see whether two drugs have similar mechanisms in treating cancer cell lines.

Figure 5 showed that the PCC between NMI and FDA-approved CNS cancer drugs were all in green (<0.8) or black (<0.0) (shown in the last line of these graphs). In addition, the exact numbers of PCC between them (Appendix A) were all lower than 0.8. Therefore, the GI_50_, TGI, and LC_50_ graphs of NMI were all different from existing CNS cancer drugs, indicating that NMI had a unique mechanism for inhibition of cancer growth. Similar results of PCC were also shown in prostate and NSCLC cancers (See Appendix A).

## 4. Discussion

Increased MAO A expression shows in glioma, prostate, and NSCLC cancers. Kushal et al. found MAO A activity was expressed at higher level in human and mouse glioma cell lines (1.29, 126.93 nmol/mg/h, respectively), but it was not detectable in normal human astrocytes [3]. Xu et al. found that MAO A levels was showed significantly higher levels in prostate cancer compared to normal prostate tissue with a 1.6 fold-change [12]. Liu et al. found that the level of MAO A expression in NSCLC tissue was higher than that in normal lung tissue. The relative mRNA expression of NSCLC tissue to non-tumor adjacent tissue is 2.5:1 [13].

Table 1 also shows decreased MAO A expression in six other cancers. However, it is important to note that MAO A expression levels are varied depending on the cell lines and subtypes of each cancer.

Yang et al. found that MAO A expression slightly decreased in colorectal cancer tumor tissue compared to normal colon tissue; MAO B expression correlated with poor prognosis in colorectal cancer [14].

MAO A expressions are decreased in breast cancer (Table 1). Similarly, Lo PK et al. found that the expression of MAO A in invasive breast cancer was lower than normal breast tissues [15]. Komatsu et al. found that MAO A was downregulated in triple-negative breast cancer (TNBC) [16].

The change in MAO A expression of leukemia depends on its subtypes. Ye et al. reported a decreased expression of MAO A in acute myeloid leukemia (AML) while Bdaiwi et al. reported an increase in MAO A expression in acute lymphoblastic leukemia (ALL) [17,18].

As shown in Figure 1, Figure 2 and Figure 3, in addition to CNS, prostate, and NSCLC cancers, NMI was effective to other cancers, including colon cancer, melanoma, ovarian cancer, renal cancer, and breast cancer.

MAO A inhibitor reduced glioma growth by activating macrophages in the microenvironment [3]. MAO A inhibitor has recently been shown to be a cancer immunotherapy like PD-1 antibody, which stimulates T-cell and inhibits cancer growth [19]. Thus, MAO A inhibitor may be an effective anticancer agent to cancers with and without increased MAO A expression.

Based on the GI_50_, TGI, LC_50_, and cumulative score of NMI, we found that NMI was more potent than FDA-approved anticancer drugs for CNS (Figure 4), prostate (Appendix A), and NSCLC cancers (Appendix A). Our results also showed NMI has a unique mechanism compared to existing anticancer drugs (Figure 5).

Further, we found the highest PCC was 0.63 (<0.8 means no correlation), indicating NMI had a unique mechanism comparing to all the anticancer drugs (Appendix A). The drug names and their mechanisms were shown in Appendix A.

To search for the NMI mechanism, PCC values of NMI and all unmarketed anticancer drugs were compared. The top 10 unmarketed drugs in all cancers in the NCI60 database with high PCC (>0.7) based on GI_50_ were shown in Appendix A. Among them, five compounds showed PCC (>0.75), suggesting that they had a similar mechanism to NMI. Two compounds (NSC 342443 and NSC 800374) had high PCC based on GI_50_, TGI, and LC_50_, respectively. NSC 342443 has been shown to inhibit transcription or translation in eukaryotic cells and inhibit protein synthesis [20]. Therefore, NMI may have similar mechanism. The mechanism of NSC 800374 is currently unknown. This approach can be used to search for potential mechanisms of NMI.

We have shown previously that NMI reduces the proliferation of angiogenesis and increases the macrophage and survival in the glioma xenograft mouse models implanted with drug resistant human glioma U251R [3]. Our previous in vivo studies using prostate cancer xenograft mouse models implanted with C4-2B prostate cancer cells showed that NMI is targeted specifically to the tumor, reducing the MAO A catalytic activity, tumor growth, and reducing the prostate-specific antigen (PSA), a biomarker for prostate cancer [5]. This study showed that NMI may be effective for other cancers and it outperformed the FDA-approved drugs for CNS, prostate, and NSCLC cancers with a unique mechanism. These findings further support that notion that NMI may be an effective therapy for a number of cancers and warrants further study and development of this drug.

Nevertheless, the NCI60 screening study is a limited in vitro cell lines study without considering the microenvironment of the tumors. The data provided a strong indication of NMI’s potency and unique mechanisms to 59 cancer cell lines representing nine different cancers. Further in vivo animal experiments using animal models, organoids, and a PDX model are necessary to develop NMI as a novel cancer treatment for these devastating cancers.

## 5. Conclusions

In conclusion, this study demonstrated the anticancer efficacy of NMI on NCI60 cancer cell lines. The results showed that NMI has higher potency to CNS, prostate, and NSCLC cell lines. NMI outperformed existing anticancer drugs for these three cancer types. The COMPARE algorithm showed that NMI had a unique mechanism to inhibit cancer growth compared to the existing drugs. These findings provided a basis for further in vivo animal studies and to develop NMI as a potential theranostic anticancer drug for a number of cancers.

## Figures and Tables

**Figure 1 brainsci-11-01318-f001:**
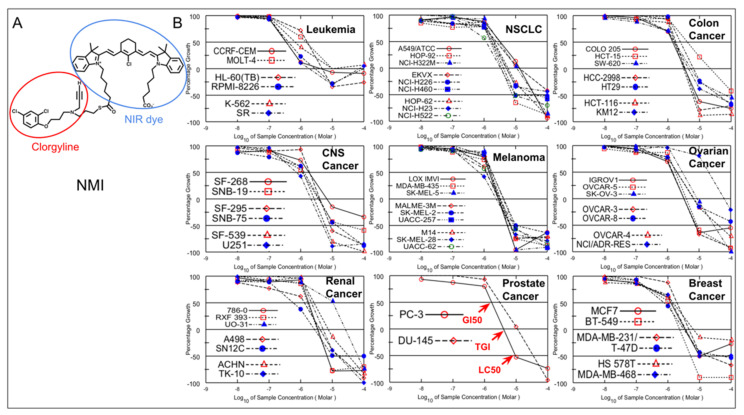
(**A**)**.** Chemical Structure of NMI. MAO A inhibitor, clorgyline (shown in red) was conjugated with MHI−148 (shown in blue) via a thioester bond. (**B**) The NMI inhibition of cell growth in 59 cell lines determined by NCI60 analysis. Five dose−response curves for NMI in 59 cell lines from nine different cancers are shown. NCI60 analysis shows NMI inhibits many cell lines in nine different cancers in NCI60 database. NMI concentration, from 10^−4^ to 10^−8^ M; 100% represents the cell growth without treatment. The 0% growth indicates no cell growth, corresponding to the number of cells at the beginning. The −100% growth means all cells are killed by treatment. GI_50_, the NMI concentration required for 50% cell growth inhibition; TGI, the concentration required for total inhibition (0%); LC_50_, the concentration required for 50% cell death.

**Figure 2 brainsci-11-01318-f002:**
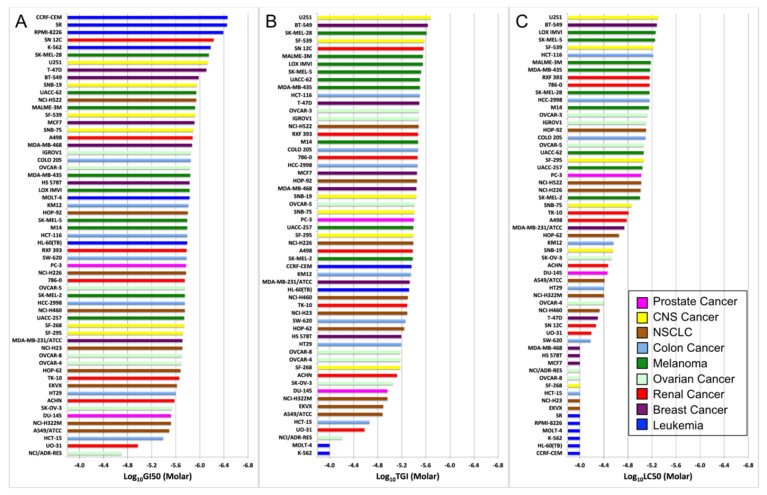
Waterfall plot of (**A**) GI_50_, (**B**) TGI, (**C**) LC_50_ of NMI. The most potent cell lines of each parameter are at the top of the plot, with the lowest molar concentration required. *x*-axis represents log concentration in GI_50_, TGI, and LC_50_ (−3.8 M to −6.8 M). Cancer types are color-coded.

**Figure 3 brainsci-11-01318-f003:**
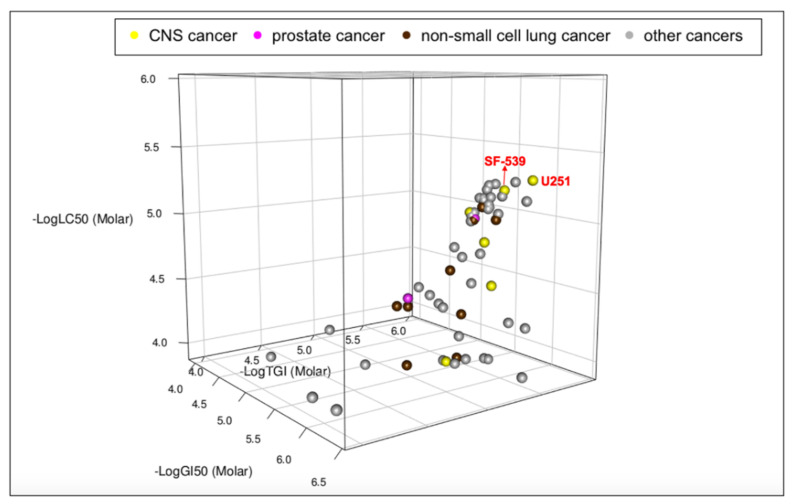
The GI_50_, TGI, and LC_50_ of NMI in a 3D scatter plot. The graph is generated by R Studio for programming. NMI was more potent in CNS, prostate, and NSCLC cancers based on the 3D scatter plot. *x*-axis is the log value of GI_50_, *y*-axis is the log value of TGI, and *z*-axis is the log value of LC_50_. The location of each dot in the plot represents the efficacy of each cell line to NMI. Cancer types are color-coded.

**Figure 4 brainsci-11-01318-f004:**
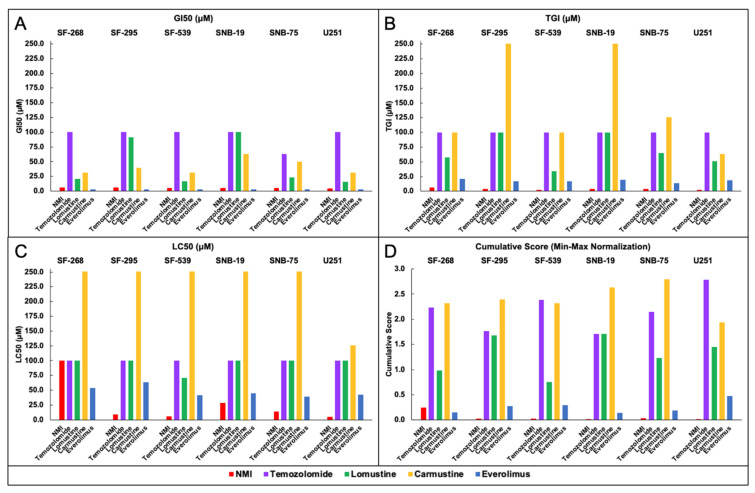
The comparison of (**A**) GI_50_, (**B**) TGI, (**C**) LC_50_, (**D**) the cumulative score of NMI to FDA-approved CNS cancer drugs. NMI shows higher potency to CNS cancer cell lines than FDA-approved CNS drugs. *y*-axis indicates the concentration of GI_50_, TGI, LC_50_ (0 to 250 μM), and cumulative score (0.0 to 3.0). Drugs are color-coded.

**Figure 5 brainsci-11-01318-f005:**
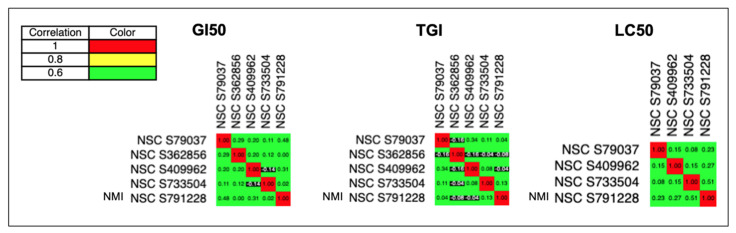
Comparison of the mechanism of NMI to FDA−approved CNS cancer drugs by COMPARE plots. NMI has unique mechanism compared to FDA approved CNS drugs. High PCC (Pearson Pairwise correlation coefficient), >0.8 shown in yellow (■), or red (■) in the matrix COMPARE figure, indicates these two drugs have similar mechanisms for cancer therapy.

**Table 1 brainsci-11-01318-t001:** The MAO A expression in tumor and normal tissues in nine different cancers used in NCI60 analysis. CNS, prostate, and NSCLC cancers showed higher MAO A expression than normal tissues; other cancers did not. The MAO A expression values are from the CRN and EMBL-EBI databases. Data are reported as mean TPM ^a^ and FPKM ^b^.

	MAO A Expression CRN			MAO A Expression EMBL-EBI		
	Normal (TPM ^a^)	Tumor (TPM)	Fold Change (Tumor/Normal)	Normal (TPM)	Tumor (TPM)	Fold Change (Tumor/Normal)
CNS Cancer	8.14 (FPKM ^b^)	10.94 (FPKM)	1.34	20 (Putamen)	31 (Glioma)	1.55
Prostate Cancer	75.9	114.28	1.51	108	192	1.78
Non-Small Cell Lung Cancer	0.28 (FPKM)	0.64 (FPKM)	2.29	No data	No data	/
Colon Cancer	169.86	43.34	0.26	199 (Transverse) 155 (Sigmoid)	62	0.31 (Transverse) 0.4 (Sigmoid)
Melanoma	42.13	0.67	0.02	64 (Suprapubic skin)	4	0.06
Ovarian Cancer	No data	4.41	/	113	14	0.12
Renal Cancer	80.63	40.78	0.51	68 (Cortex of kidney)	60	0.88
Breast Cancer	103.91	12.26	0.12	102	4	0.04
Leukemia	No data	0.21	/	4 (Blood)	No data	/

^a^ TPM: transcripts per 1,000,000 mapped reads, a unit of the proportion of transcripts in mRNA; ^b^ FPKM: fragments per kilobase of exon per million reads, a normalized fragment of DNA divided by the total length of all exons in the gene (or transcript).

## Data Availability

The data presented in this study are available within the article. For any further request contact the corresponding author.

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
