# Peer review of "Near-Infrared MAO A Inhibitor (NMI) Outperformed FDA-Approved Chemotherapeutic Agents in Brain and Other Cancers: A Bioinformatic Analysis of NCI60 Screening Data"

_brainsci, 2021, doi:10.3390/brainsci11101318_

Round 1
Reviewer 1 Report
This excellent work explores the toxicity of NMI by screening with the NCI 60 panel. NMI had a strong toxicity on many cell lines. Intriguingly, the effect was stronger than approved drugs. NMI have the advantage, that they might be used as theragnostics and therefore also be used for imaging.
All resuts are sound and the conclusions drawn are convincing. This work will be of high interest to your readers. The results presented here will be the base for further experiments, i.e. animal experiments before early clinical trials may be started.
The linguistic style is very good, making the manuscript easy to read and understand.
Author Response
Thank you for your time to review. Thank you for considering our manuscript as excellent, sound, convincing, and interesting. We appreciate that you think our work will be of high interest and easy to understand for readers.
Reviewer 2 Report
The following manuscript was reviewed. brainsci-1296779 ‘Near-infrared MAO A Inhibitor outperformed FDA-approved chemotherapeutic agents in brain and other cancers: a bioinformatic analysis of NCI60 screening data’
This is a very interesting report and if improved could have a wider interest to many in the brain sciences fields. The major concerns this reviewer has is the lack of detailed information on approach and methods. While it may be good for those knowledgeable in this particular field of bioinformatic analysis, it would be nice if more detailed information was provided. Forgive my lack of knowledge with using this screening data, was the NMI sent to NCI to perform the studies? It would be nice to have information on the conjugation chemistry - I did look over previous work and could not find the details on the chemistry. Was the non-conjugated MAO A inhibitor tested as well? The approach for the literature and database search from CRN and EMBL-EBI is not clear and should be presented in materials and methods. The link to the NCI-60 screening methodology is good, however could the authors summarise for this report?
Abbreviations should be defined in text
Page 4 line 1 of top paragraph – space needed between foldin. Also, on page 4, it seems like some of the discussion in this section would be better placed in discussion.
Figure 2 – labels seem out of focus
Again, this next set of questions may be due to my inexperience with this type of analysis but I am not sure I understand the following:
“To further investigate the potency of NMI to these 3 cancers, NCI60 screening data of GI50, TGI and LC50 values for NMI were arranged and made into waterfall plots (Figure 2). The drug efficacy of NMI was determined by the values of these 3 parameters, the higher drug efficacy required the lower concentration in GI50, TGI, and LC50. Therefore, the cell line at the top of the figure was the most potent, and the potency of the following cell lines decreased progressively.”
Do you mean 3 cancer types?
What is meant by the cell line was the most potent?
Page 6 - seems like some of the detailed and necessary information would be better placed in materials and methods
Figure 4 – curious to why the data isn’t presented like figure 2. Also, were these experiments conducted at the same time or does the data represent means or averages of different experiments? Not clear on this aspect. A discussion concerning the caveats or limitations of this type of study would be beneficial.
Author Response
Please see the attachment.

This manuscript is a resubmission of an earlier submission. The following is a list of the peer review reports and author responses from that submission.